# Risk of Cancer in Family Members of Patients with Lynch-Like Syndrome

**DOI:** 10.3390/cancers12082225

**Published:** 2020-08-09

**Authors:** María Dolores Picó, Ana Beatriz Sánchez-Heras, Adela Castillejo, Mar Giner-Calabuig, Miren Alustiza, Ariadna Sánchez, Leticia Moreira, María Pellise, Antoni Castells, Gemma Llort, Carmen Yagüe, Teresa Ramon y Cajal, Alexandra Gisbert-Beamud, Joaquin Cubiella, Laura Rivas, Maite Herraiz, Catalina Garau, Inmaculada Salces, Marta Carrillo-Palau, Luis Bujanda, Adriá López-Fernández, Cristina Alvarez-Urturi, María Jesús López, Cristina Alenda, Pedro Zapater, Francisco Javier Lacueva, Francesc Balaguer, Jose-Luis Soto, Óscar Murcia, Rodrigo Jover

**Affiliations:** 1Servicio de Medicina Digestiva, Hospital General Universitario de Elche, Elche, 03203 Alicante, Spain; madopisa@hotmail.com; 2Servicio de Oncología Médica, Hospital General Universitario de Elche, Elche, 03203 Alicante, Spain; zrtbns@gmail.com; 3Unidad de Genética Molecular, Hospital General Universitario de Elche, 03203 Alicante, Spain; castillejo.ade@gmail.com (A.C.); sotomartinez.jl@gmail.com (J.-L.S.); 4Servicio de Medicina Digestiva, Hospital General Universitario de Alicante, Instituto de Investigación Sanitaria ISABIAL, 03010 Alicante, Spain; maretagc@gmail.com (M.G.-C.); mirenalustiza92@gmail.com (M.A.); 5Servicio de Medicina Digestiva, Hospital Clínic de Barcelona, IDIBAPS, CIBERehd. University of Barcelona, 08036 Barcelona, Spain; asanchezg@clinic.cat (A.S.); LMOREIRA@clinic.cat (L.M.); mpellise@clinic.cat (M.P.); CASTELLS@clinic.cat (A.C.); fprunes@clinic.cat (F.B.); 6Servicio de Oncología Médica, Hospital Universitari Parc Taulí, Sabadell, Consorci Sanitari de Terrassa, Terrassa, 08208 Barcelona, Spain; llortgemma@gmail.com (G.L.); cyague@cst.cat (C.Y.); 7Servicio de Medicina Digestiva, Hospital de la Santa Creu i Sant Pau, 08041 Barcelona, Spain; tramon@santpau.cat (T.R.yC.); AGisbert@santpau.cat (A.G.-B.); 8Servicio de Medicina Digestiva, Complexo Hospitalario Universitario de Ourense, Instituto de Investigación Sanitaria Galicia Sur, CIBERehd, 32005 Ourense, Spain; joaquincubiella@gmail.com (J.C.); Laura.Rivas.Moral@sergas.es (L.R.); 9Servicio de Medicina Digestiva, Clínica Universidad de Navarra, 31008 Navarra, Spain; mherraizb@unav.es; 10Servicio de Medicina Digestiva, Hospital de Son Llàtzer, 07198 Palma de Mallorca, Spain; cgarau@hsll.es; 11Servicio de Medicina Digestiva, Hospital 12 de Octubre, 28041 Madrid, Spain; isalces@live.com; 12Servicio de Medicina Digestiva, Hospital Universitario de Canarias, 38320 Tenerife, Spain; martacarry@yahoo.es; 13Servicio de Medicina Digestiva, Hospital Donostia/Instituto Biodonostia, Centro de Investigación Biomédica en Red de Enfermedades Hepáticas y Digestivas (CIBERehd), Universidad del País Vasco (UPV/EHU), 20014 San Sebastián, Spain; luis.bujandafernandezdepierola@osakidetza.eus; 14Unidad de Alto Riesgo y Prevención del Cáncer, Hospital Universitario Vall d’Hebron, 08035 Barcelona, Spain; adlopez@vhio.net; 15Servicio de Medicina Digestiva, Hospital del Mar, 08003 Barcelona, Spain; ACAlvarez@parcdesalutmar.cat; 16Servicio de Medicina Digestiva, Hospital Universitario Marqués de Valdecilla, 39008 Santandercity, Spain; mariajesuslopez33@gmail.com; 17Servicio de Anatomía Patológica, Hospital General Universitario de Alicante, Instituto de Investigación Sanitaria ISABIAL, 03010 Alicante, Spain; alenda.cris@gmail.com; 18Servicio de Farmacología Clínica, Hospital General Universitario de Alicante, ISABIAL, CIBERehd, IDiBE, UMH, 03010 Alicante, Spain; zapater.p@gmail.com; 19Servicio de Cirugía general, Hospital General Universitario de Elche, Elche, 03203 Alicante, Spain; fj.lacueva@umh.es

**Keywords:** colorectal cancer, risk, genetic, surveillance

## Abstract

Lynch syndrome (LS) is a common cause of hereditary colorectal cancer (CRC). Some CRC patients develop mismatch repair deficiency without germline pathogenic mutation, known as Lynch-like syndrome (LLS). We compared the risk of CRC in first-degree relatives (FDRs) in LLS and LS patients. LLS was diagnosed when tumors showed immunohistochemical loss of MSH2, MSH6, and PMS2; or loss of MLH1 with *BRAF* wild type; and/or no *MLH1* methylation and absence of pathogenic mutation in these genes. CRC and other LS-related neoplasms were followed in patients diagnosed with LS and LLS and among their FDRs. Standardized incidence ratios (SIRs) were calculated for CRC and other neoplasms associated with LS among FDRs of LS and LLS patients. In total, 205 LS (1205 FDRs) and 131 LLS families (698 FDRs) had complete pedigrees. FDRs of patients with LLS had a high incidence of CRC (SIR, 2.08; 95% confidence interval (CI), 1.56–2.71), which was significantly lower than that in FDRs of patients with LS (SIR, 4.25; 95% CI, 3.67–4.90; *p* < 0.001). The risk of developing other neoplasms associated with LS also increased among FDR of LLS patients (SIR, 2.04; 95% CI, 1.44–2.80) but was lower than that among FDR of patients with LS (SIR, 5.01, 95% CI, 4.26–5.84; *p* < 0.001). FDRs with LLS have an increased risk of developing CRC as well as LS-related neoplasms, although this risk is lower than that of families with LS. Thus, their management should take into account this increased risk.

## 1. Introduction

Lynch syndrome (LS) is the most frequent cause of hereditary colorectal cancer (CRC) [1]. It is characterized by the presence of pathogenic mutations in DNA mismatch (MMR) repair genes (*MLH1, MSH2, MSH6, PMS2*) or *EpCAM* (epithelial cell adhesion molecule), with an autosomal dominant inheritance [2]. It is mainly characterized by the high risk of developing CRC and endometrial cancer, as well as other neoplasms, namely of the ovaries, urinary tract, stomach, small intestine, pancreas, biliary tract, skin, and brain. Implementation of Jerusalem guidelines, which recommend conducting MMR proteins immunohistochemistry (IHC) in all CRC or endometrial cancers in patients younger than 70 years old [3] or universal screening, has increased the number of patients diagnosed with LS but has also revealed a large number of patients (~30%) who present with tumor microsatellite instability (MSI) or loss-of-expression MMR proteins but without evidence of germline pathogenic mutation in these genes [4]. This condition is known as Lynch-like syndrome (LLS). In this context, it is important to adequately address the existence of variants of unknown significance (VUS) following international guidelines for classification [5,6,7,8]. Inadequately classified VUS in MMR genes could also be a cause of LLS.

Currently, it is unclear what should be the type of follow-up in these patients and their relatives. A previous study by our group [9] showed that the risk of CRC was lower in families of patients with LLS compared to the relatives of patients with LS but was higher than that found in relatives of patients with sporadic cases. No increase in the risk of LS-related neoplasms was found in this cohort. However, this study had some limitations, mainly the low number of families with LLS cases, which may have led to selection bias in the cohort.

The goal of this study was to evaluate, in a large cohort, the risk of developing CRC and other LS-related neoplasms among the family members of patients diagnosed with LLS compared with a cohort of LS families. We also compared the clinical and pathological characteristics of CRC patients with LS and LLS.

## 2. Material and Methods

### 2.1. Patients and Data Collection

Data were extracted from a descriptive, observational, multicenter, nationwide registry (EPICOLON III) on familial CRC, involving 25 Spanish hospitals [10]. We studied a cohort of patients with CRC and LS (with confirmed germline pathogenic mutation in mismatch repair (MMR) genes) and patients with CRC and LLS. LLS is defined by the presence of high MSI (MSI-H) and/or loss of expression of MMR proteins as determined by immunohistochemistry in CRC patients, without *MLH1* promoter hypermethylation or evidence of germline pathogenic mutation in the DNA MMR genes or epithelial cell adhesion molecule (*EpCAM*). IHC and/or MSI study of the tumors was performed because of fulfillment of Amsterdam criteria and/or revised Bethesda guidelines [11], or because of universal molecular screening for LS [3]. Patients were investigated according to common protocols [12], and were included since November 2017. If any pathogenic mutation in MMR genes was found, genetic testing was expanded among the first-degree relatives (FDRs) of the index case. We followed up with these patients and their relatives until July 2019, with the purpose of learning about new cases of CRC and other LS-associated tumors between these patients and their relatives.

Data on demographic variables, study of CRC, and family history were obtained from the national registry EPICOLON III (www.epicolon.es). All colorectal carcinomas were analyzed for grade, location, Tumor, lymph Nodes, Metastasis (TNM) stage, and for histopathologic features, such as mucinous cell histology, vascular invasion, and tumor-infiltrating lymphocytes, based on morphologic assessment. In all cases, the family history was collected through the cancer pedigrees that included at least one generation backward and forward to the index case, as previously described [9]. All of the patients provided written informed consent. The study was approved by the institutional review boards of the participating hospitals (date: 23 June 2013, ethic code: PI2013073).

### 2.2. MSI, IHC Staining, and Detection of Germline Mutations

MSI and/or IHC analysis was performed in all the CRC patients included in this cohort. MSI status [13,14] and IHC analysis of MLH1, MSH2, MSH6, and PMS2 was performed in formalin-fixed paraffin-embedded tumor tissue, as previously described [15]. In patients with loss of MLH1, the methylation of *MLH1* and/or somatic V600E *BRAF* mutation status was analyzed. *MLH1* methylation analysis was performed using methylation-specific multiplex ligation-dependent probe amplification (MS-MLPA) using the SALSA MS-MLPA kit ME011 Mismatch Repair Genes (MRC-Holland, Amsterdam, The Netherlands) [16]. The V600E *BRAF* mutation was detected by real-time PCR (ABI Prism 7500, Applied Biosystems, Foster City, CA, USA) using specific TaqMan probes and allelic discrimination software as previously described [17].

Germline mutation analysis was performed in accordance with the results of IHC analysis as previously described [4]. The selection of patients and genes for germline mutation analysis was based on IHC and MSI results. Accordingly, *MLH1* and *MSH2* mutational analysis was performed in all tumors with MLH1 and MSH2-negative staining. *MSH6* germline analysis was done in patients with an isolated lack of MSH6 expression or combined lack of MSH2 and MSH6 not showing *MSH2* mutation. Genetic testing for *PMS2* was performed in an isolated loss of PMS2 expression tumors. Patients with loss of MSH2 expression with no mutation detected were analyzed for *EpCAM* rearrangements. In patients showing MSI and whose IHC analysis could not be performed, genetic testing of all four MMR genes (*MLH1, MSH2, MSH6*, and *PMS2*) was carried out. DNA sequencing was performed to characterize the deletion breakpoints. Germline mutation studies were performed on genomic DNA isolated from peripheral blood leucocytes or from non-tumor colon tissues. Detection of point mutations was conducted using PCR and direct sequencing of the whole coding sequence and intron-exon boundaries of each gene. Large rearrangements (deletions and insertions) were tested by MLPA (MLPA Kits P003: *MLH1-MSH2*; P248 (*MLH1-MSH2* confirmation), P008 (*PMS2-MSH6*), and P072 (*EpCAM*) MRC-HOLLAND, Amsterdam, The Netherlands) following the manufacturer’s protocol. Large rearrangements (deletions and insertions) were tested using MLPA according to the manufacturer’s protocol. The results of the genetic analysis were interpreted based on the ACMG Recommendations for Standards for Interpretation of Sequence Variations (2000) and the InSIGHT database [5,7].

### 2.3. Calculation of Standardized Incidence Ratio

To analyze the family risk of developing CRC or other tumors associated with LS, we included information relative to the FDRs of patients with CRC and LS, regardless of whether or not they were carriers of the mutation, or LLS only in cases with complete pedigrees and information about the ages of all family members, including relatives without cancer. We calculated the standardized incidence ratio (SIR) as the ratio of the observed to expected number of cases diagnosed in the families at the time of inclusion in the EPICOLON III cohort. The expected number of cases was calculated as the sum of the products of the number of person-years for each 5-year age/sex group and the corresponding age/sex-specific incidence rates in Spanish regional registries [18]. The confidence limits were based on Byar’s approximation of the exact Poisson distribution, which is accurate even with small numbers [19]. The index case was excluded for the analysis of family history at the time of diagnosis.

We also prospectively analyzed the risk of developing CRC or other neoplasms associated with LS during follow-up in family members of patients with LS and LLS. We considered as LS-related tumors those from the endometrium, ovaries, urinary tract, stomach, small intestine, pancreas, biliary tract, skin, and brain. We followed up with new cases of these tumors in the index patients and their FDRs that appeared since the moment of diagnosis of the index case until July 2019. We included information regarding FDRs who had complete information about the surveillance of FDRs (age, history of neoplasia), excluding the rest due to the absence of information.

The pedigrees were updated by asking patients and/or relatives about new cases of cancer after diagnosis of the index case. We included the index case for this analysis, and the appearance of metachronous CRC or a new case of non-colorectal LS-related tumor in the index case was considered a new case in the family.

### 2.4. Statistical Analysis

Regarding the descriptive analysis, the qualitative variables are presented as percentages. The continuous quantitative variables are described from the mean and the standard deviation (SD) or from the median and the interquartile range (IQR), depending on whether or not they followed a normal distribution. For analysis of the association between qualitative variables, the chi-square test was used, followed by the Fisher’s exact test and the Student’s *t*-test or the Mann–Whitney U test for the quantitative variables. For the contrast of the hypotheses described above, a confidence level (*p*) < 0.05 was used. The incidence rates of new cases of affected family members of CRC and of non-colonic neoplasms associated with LS were calculated. Next, the incidence rates of CRC or other neoplasms associated with LS were standardized by sex and age using the indirect method. The follow-up times (person-year) for each family member at risk were calculated from 20 years of age to the time of diagnosis of the earliest cancer, age at death, or end of follow-up. The SIR was calculated from the quotient between the observed cases and those expected among the relatives. The number of expected cases was calculated from the sum of the follow-up times (person-year) for age groups with a 5-year interval, and by sex and the corresponding standard incidence rate. Statistical analysis was performed with SPSS software (SPSS 19.0, Chicago, IL, USA).

## 3. Results

### 3.1. Clinical and Pathology Differences in CRC between LS and LLS

Through July 2019, we included 160 patients diagnosed with CRC who met the diagnostic criteria of LLS and 286 patients diagnosed with CRC and LS. A flow chart of patients and FDR included can be seen in Figure 1. The characteristics of both populations are shown in Table 1. Patients with LS were younger at diagnosis of CRC (LS, 48.1 (SD 12.9) vs. LLS, 54.9 (SD 14.2); *p* = 0.01). We also observed significant differences regarding a higher percentage of patients in the LS group who met the criteria of Amsterdam and/or Bethesda (Amsterdam I and II present in 72.1% in the LS group vs. 11.2% in the LLS group, *p* = 0.00; Bethesda guidelines: 84.2% in the LS group vs. 64.4% in the LLS group; *p* = 0.00). Patients with LLS showed more frequently loss of MLH1/PMS2 expression than LS patients (48.1% vs. 34.3%; *p* = 0.004) and less frequently loss of MSH2/MSH6 (26.9% vs. 42.6%; *p* = 0.001). Regarding patients with LS, 35.3% presented with a mutation in *MLH1*, 41.3% in *MSH2*, 14.7% in *MSH6*, 5.2% in *PMS2*, and 3.5% in *EpCAM*.

We did not find relevant differences regarding the location of the CRC or TNM stage, but when we analyzed the pathology features, we only found a higher percentage of lymphocytic infiltration in the CRC of patients with LLS compared to patients with LS (23.1% in LLS vs. 14.3% in LS, *p* = 0.04). However, we did not find relevant differences regarding vascular invasion, grade of differentiation, or the presence of mucinous tumor (Table 1). On the other hand, significant differences were also observed in relation to the personal history of CRC or other neoplasms associated with LS. Synchronous CRC was present in the LS group in 9.1% of them and in the LLS group was present in 1.3% (*p* = 0.001). Metachronous CRC developed in 12.9% of the LS group versus 3.1% of the LLS group (*p* = 0.001), and 29.4% of patients in the LS group had a personal history of other tumors associated with LS compared to 3.1% of patients in the LLS group (*p* = 0.00; Table 1). Examples of the pedigrees of LS and LLS families can be found in Figure 2.

### 3.2. Risk of CRC in FDRs of LS and LLS Patients

As previously explained in the methods section, for the analysis of family risk of developing CRC or other neoplasms associated with LS, only families with complete information on age and history of neoplasms were included. A total of 205 out of 286 families with LS were included. Regarding LLS patients, this information was obtained in 131 out of 160 families. Finally, a total of 1903 FDRs were included, namely, 1205 in the LS group and 698 in the LLS group (Figure 1). When we analyzed the SIR for CRC between both groups, there was an increased risk of CRC in both LS and LLS families, with the SIR being significantly higher for the relatives of patients with LS compared to those of patients with LLS (SIR for CRC in LS, 4.25; 95% confidence interval (CI), 3.67–4.90 vs. SIR for CRC in LLS, 2.08; 95% CI, 1.56–2.71; *p* < 0.001). The same was observed for LS-associated neoplasms (other than CRC); in both cases, there was an increased risk of these neoplasms, which was also higher in LS (SIR, 5.01; 95% CI, 4.26–5.84) than in LLS (SIR, 2.04; 95% CI, 1.44–2.80; *p* < 0.001) (Table 2). The location of non-CRC neoplasms can be seen in Table 3, a higher frequency of endometrial cancer was detected in LS families (48.4% in LS group vs. 20.0% in the LLS group; *p* = 0.001). However, the frequency of pancreatic neoplasms was higher in the LLS group (3.3% in LS group vs. 15.0% in the LLS group; *p* = 0.003).

We also conducted a prospective follow-up from the diagnosis of patients with LS or LLS analyzing the new cases of CRC or other neoplasms associated with LS with a median of 3 years (IQR, 1–6) of prospective follow-up. In this analysis, we included information regarding LS patients and their FDRs, and LLS patients and their FDRs. From a total of 1126 FDR of LS patients, 22 cases of CRC (1.9%) were observed during follow-up in front of only 3 cases (0.50%) of the total 587 patients in the group of LLS and FDRs. On the other hand, there were 23 cases that developed other neoplasms associated with LS (2%) during the follow-up between LS patients and their FDRs compared to two new cases in the group of LLS patients and their FDRs (0.3%) (Table 4). Figure 3 shows the Kaplan–Meier chart of this prospective follow-up, showing that LS families had a higher incidence of new cases of LS-related cancers than LLS families (log rank, 0.0001).

## 4. Discussion

The primary finding of this study was the increased risk of CRC among FDRs of patients with LLS, which was lower than that found for LS families but significantly higher than expected. In addition, the risk of LS-related cancer was also increased in LLS families. Our results confirm previous preliminary results, extending the risk of cancer to LS-related neoplasms, and highlights the need for specific surveillance for these patients and their FDRs. These results also emphasize the need for molecular tools that distinguish between truly hereditary and sporadic cases of LLS.

With the implementation of universal tumor screening for LS in all patients with CRC [3], an increase in the number of patients with LLS has been shown. These patients represent a heterogeneous population where sporadic and truly LS cases are mixed [10]. This fact makes the adequate follow-up of these patients and their FDR difficult as, in the majority of cases, they are not appropriately diagnosed and classified. There are different explanations for the finding of LLS [10]. The first potential explanation that should be ruled out is false-positive cases of MSI or IHC. In cases of LLS, the reassurance of these results and performance of both is mandatory, and it has been described that up to 19% of cases are due to false-positive screening results [20]. Additionally, VUS in MMR genes should be adequately addressed. A major challenge in the diagnosis and management of LLS is the frequent occurrence of VUS in these genes. Depending on the gene, about 1/5th to 1/3rd of DNA sequence variants identified during the course of LS clinical testing are of uncertain significance [21]. Recently, the International Society for Gastrointestinal Hereditary Tumors (InSiGHT) developed criteria for the interpretation of the MMR gene variants, with the aim to improve the clinical utility of genetic testing for LS. Adequate classification of VUS following the InSiGHT criteria is another important step in the correct diagnosis of LLS cases [7]. Once confirmed that this is an LLS case, there are two possibilities: Cases can be due to germline mutations that have not been detected, or can be sporadic cases due to biallelic somatic inactivation of MMR genes or other genes related to the MMR system. This inactivation can be induced by double hit somatic mutations in these genes, inactivation due to loss of heterozygosity, or a combination of both. Additionally, mosaicism in MMR genes can be a potential cause for that inactivation. Different authors have proposed the investigation of double somatic mutations in MMR and other genes that could explain sporadic CRC cases with LLS, and findings of somatic mutations have been found in a variable proportion of cases, ranging from 22% to 69% [20,22,23,24,25]. Although, in recent years, there has been an increase in the use of multigene panel testing in LLS, the best diagnostic strategy to perform in patients with LLS has not been established, with no specific methodology uniformly recommended. Moreover, the addition of somatic mutations to the diagnostic algorithm of LS has not been validated in research studies. Finally, a relationship between these somatic mutations and germline inactivation of still unknown genes related to MMR deficiency has not yet been fully ruled out, and only a germline exome approach or a clinical follow-up validation will confirm the sporadic behavior of these LLS tumors with somatic mutations [10].

Additionally, as previously reported, there are no clinical or histopathological characteristics that can help differentiate between potentially hereditary or sporadic cases [10,26]; neither family history nor age at diagnosis can differentiate between cases, and with no accepted and validated molecular tool for making this distinction, LLS patients and families must be managed as a group. Moreover, in previous studies that also used somatic mutations to classify LLS patients as hereditary or sporadic, authors did not find any clinical or pathological characteristics that were able to differentiate between the two populations [27,28].

In this study, which included a large cohort of patients with LS and LLS diagnosed with CRC, we compared the clinical and molecular characteristics of these patients. Significant differences were observed regarding the age of CRC diagnosis, which was higher in patients with LLS, whereas a higher percentage of compliance of Amsterdam and Bethesda criteria was observed in patients with LS. It is worth mentioning, with respect to the histological characteristics, that a greater significant percentage of lymphocytic infiltration occurred in patients with LLS, although this finding has been described as a typical characteristic of MMR-deficient tumors, irrespective of their hereditary or sporadic origin. In addition, the increase in metachronous and synchronous CRC present in patients with LS was also significant, as well as the development of other malignancies associated with LS. On the other hand, as mentioned above, although the risk of CRC or other neoplasms associated with LS among family members of patients with LLS was increased, it was still lower than the risk presented by family members of patients with LS. Similar results were observed in the study previously conducted by Rodriguez-Soler [9] and also by Win et al. [29]. Our results support establishing surveillance for the FDRs of patients with LLS. Cancers found in this follow-up were significantly more frequent in LS than in LLS families; however, the increased risk found in LLS still justifies this surveillance. 

The main limitation of this study is the lack of molecular information about somatic mutations in the LLS cases. Therefore, a more accurate estimate of cancer risk cannot be performed among the subgroups of LLS, namely patients with putative germline mutations that cannot be detected due to the limitations of the technology and those with the presence of somatic biallelic inactivation and probably sporadic origin. Other limitations of our study were the possibility of underreporting or misreporting cancers because our information was not always confirmed with objective clinical and pathologic data. However, we believe this limitation is minor because it would affect the LLS group to the same extent as the other groups. Another limitation of our study was the lack of a homogeneous follow-up for FDRs of patients with LLS between the participant centers, which could have led to less follow-up for LLS than LS patients and could potentially explain the reduction in the number of incident cases between LLS families. Furthermore, the prospective follow-up time was too short, and longer periods could better demonstrate the efficacy of surveillance in these LLS families. It is also possible that the lack of a clear diagnosis decreased the adherence to follow-up in the LLS families [30]. On the other hand, the strength of our study was the large cohort of LS and LLS patients and families with carefully updated pedigrees. This approach provided robustness to our data in terms of potential applicability to general practice.

In summary, in this large cohort study, we confirmed that the FDRs of patients with LLS have a higher risk of developing CRC and other LS-related neoplasms, although this risk was significantly lower than that found in LS families. These results highlight the need to develop molecular tools able to clearly separate truly hereditary from sporadic cases. Meanwhile, LLS should be considered a high-risk disease and patients and their FDRs should be followed up accordingly. Thus, because, as a group, the risk of CRC in FDR of patients with LLS was higher than expected, we recommend screening for these relatives, as well as gynecological examination, until the hereditary origin of the tumor can be effectively ruled out.

## 5. Conclusions

First-degree relatives of Lynch-like syndrome have an increased risk of developing LS-related neoplasms. These data support the need for applying preventive measures for these patients and their relatives.

## Figures and Tables

**Figure 1 cancers-12-02225-f001:**
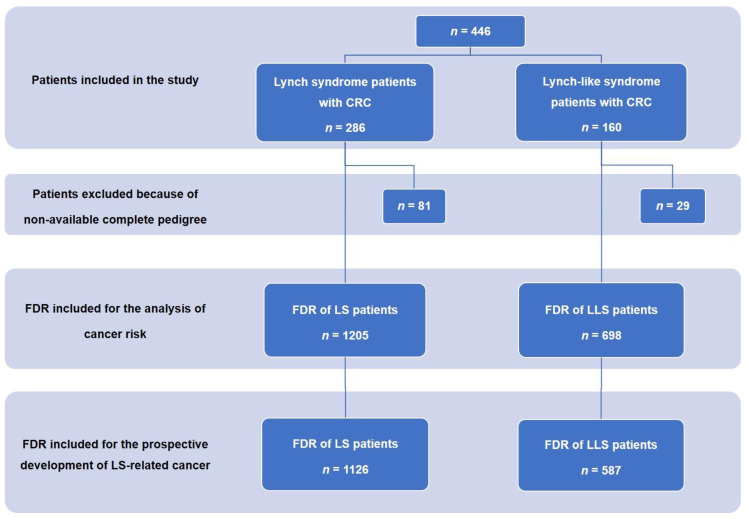
Flow diagram of patients and FDR included. FDR, first-degree relatives; LS, Lynch syndrome; LLS, Lynch-like syndrome; CRC, colorectal cancer.

**Figure 2 cancers-12-02225-f002:**
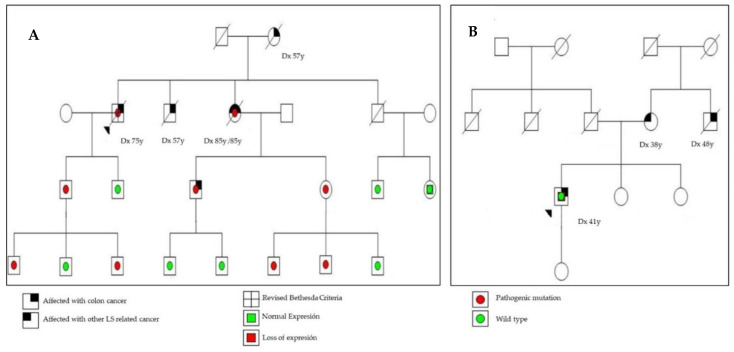
Examples of pedigrees of Lynch syndrome (**A**) and Lynch-like syndrome (**B**) families. LS, Lynch syndrome; Dx, diagnosis, Y, years.

**Figure 3 cancers-12-02225-f003:**
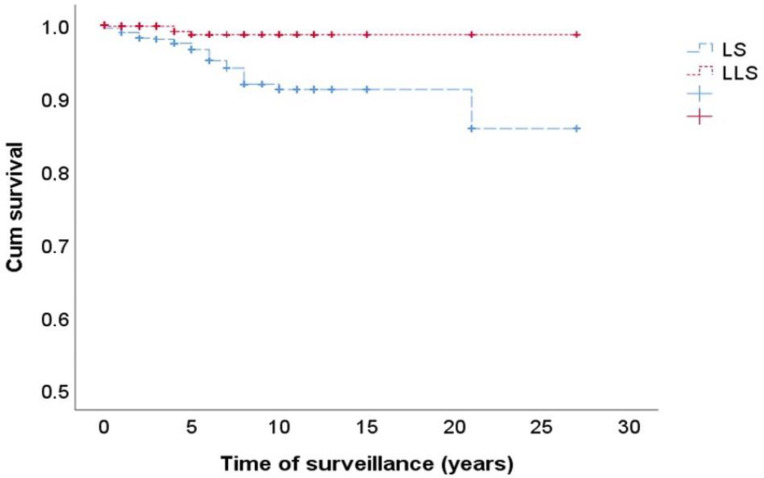
Kaplan–Meier curve showing the risk incidence of CRC and other LS-associated tumors during follow-up in the FDRs of patients with LS and LLS. Log Rank, *p* = 0.000.

**Table 1 cancers-12-02225-t001:** Characteristics of patients with LS and LLS.

Characteristics of Patients	LS, *n* = 286	LLS, *n* = 160
Female sex	132 (46.1%)	66 (41.3%)
Age at CRC diagnosis, median (SD)	48.1 (SD 12.9)	54.9 (SD 14.2) *
Reason for IHC		
Amsterdam I and II criteria	206 (72.1%)	18 (11.2%) *
Revised Bethesda guidelines	234 (84.2%)	103 (64.4%) *
Immunohistochemistry (IHC), *n* (%)		
Loss of MLH1 and PMS2	98 (34.3)	77 (48.1) *
Loss of MSH2 and MSH6	122 (42.6)	43 (26.9) *
Isolated loss of MSH6	40 (14)	20 (12.5)
Isolated loss of PMS2	18 (6.3)	14 (8.8)
IHC non-available; MSI-H	8 (2.8)	6 (3.7)
Location		
Right colon	165 (62.7%)	89 (61.4%)
Left colon and rectum	98 (37.3%)	56 (38.6%)
TNM		
Stage I and II	159 (55.6%)	80 (60.1%)
Histology		
Poor differentiation	50 (17.5%)	33 (20.6%)
Lymphocytic infiltration	41 (14.3%)	37 (23.1%) *
Mucinous tumor	88 (30.8%)	46 (28.7%)
Vascular invasion	35 (12.2%)	18 (11.3%)
Personal history		
metachronous CRC	37 (12.9%)	5 (3.1%) *
synchronous CRC	26 (9.1%)	2 (1.3%) *
non-CRC LS tumor	84 (29.4%)	5 (3.1%) *

* *p* < 0.05. SD, standard deviation; CRC, colorectal cancer; LS, Lynch syndrome; LLS, Lynch-like syndrome; MSI-H, microsatellite instability-high, TNM, Tumor, lymph Nodes, Metastasis

**Table 2 cancers-12-02225-t002:** SIR of CRC or other neoplasms associated with LS in first-degree relatives of patients with LLS compared with FDR of patient with LS.

Neoplasms Associated with LS	LS *n* = 1205		LLS *n* = 698		*p* Value
	Number of Tumors	SIR (95% CI)	Number of Tumors	SIR (95% CI)	
CRC	191	4.25 (3.67–4.90)	54	2.08 (1.56–2.71)	0.0000
Non-CRC LS-associated tumor	161	5.01 (4.26–5.84)	38	2.04 (1.44–2.80)	0.0000
TOTAL	352	4.57 (4.10–5.07)	92	2.06 (1.66–2.53)	0.0000

*p* < 0.05. SIR, standardized incidence ratio; LS, Lynch syndrome; LLS, Lynch-like syndrome; CRC, colorectal cancer; CI, confidence interval; FDR, first-degree relatives.

**Table 3 cancers-12-02225-t003:** Non-CRC LS-associated tumors in LS and LLS patients and their FDR.

Location of the Tumors	LS Families	LLS Families	*p* Value
	Number of Tumors: 184	Number of Tumors: 40	
Ovary	20 (10.9%)	5 (12.5%)	n.s.
Endometrium	89 (48.4%)	8 (20.0%)	0.001
Pancreas	6 (3.3%)	6 (15.0%)	0.003
Stomach	37 (20.1%)	11 (27.5%)	n.s.
Urinary tract	10 (5.4%)	4 (10%)	n.s.
Skin	3 (1.6%)	2 (5%)	n.s.
Small intestine	5 (2.7%)	0 (0%)	n.s.
Brain	6 (3.3%)	2 (5%)	n.s.
Biliary tract	8 (4.3%)	2 (5%)	n.s.

*p* < 0.05. LS, Lynch syndrome; LLS, Lynch-like syndrome; CRC, colorectal cancer; n.s., no significative.

**Table 4 cancers-12-02225-t004:** Prospective appearance of new cases of cancer between LS patients and their FDR and LLS patients and their FDR.

Neoplasms Associated with LS	LS *n* = 1126	LLS *n* = 587	*p* Value
	No. of Tumors	No. of Tumors	
CRC	22 (1.9%)	3 (0.5%)	0.019
Non-CRC LS-associated tumor	23 (2%)	2 (0.3%)	0.006
TOTAL	45 (3.9%)	5 (0.8%)	0.000

*p* < 0.05. LS, Lynch syndrome; LLS, Lynch-like syndrome; CRC, colorectal cancer; FDR, first-degree relatives.

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
