# Peer review of "Risk of Cancer in Family Members of Patients with Lynch-Like Syndrome"

_cancers, 2020, doi:10.3390/cancers12082225_

Round 1

Reviewer 1 Report

The work is well written and the conclusions agree with the results.

However, there are improvements to be made. The introduction must certainly be expanded by referring also to all the variants of uncertain meaning that are usually identified in the MMR genes and which instead could be the cause of the MSI-H phenotype identified in LLS patients. In the results, in the table 1, the column referring to mutated MMR genes in LS patients could be eliminated also because it is not possible to make a comparison with the other cohort. In addition, a reference to VUS eventually identified in MMR genes among LLS patients could be added. In this regard, even in the discussions it would be interesting to consider this aspect (VUS) and discuss the topic.

The references must be updated 

Minor revision

In the table 1, the IHC results column in LLS patients should be shifted higher.

Reviewer 2 Report

Picó et al. compared the risk of having mainly colorectal cancer (CRC) in first degree relatives of Lynch syndrome (LS) and Lynch like syndrome (LLS). The idea of finding a molecular analysis to better clarify LLS is very welcome and important. This manuscript is interesting, easy reading and brings important surveillance suggestion for LLS patients. However, a few clarification suggestion can be found. First, it is not clear for me how the testing principle of LS family members is conducting in Spain. It would be nice if Authors could briefly describe this. I mean if an index case is diagnosed in LS is it automatically clear that all family members (or only first degree members or only children) of index case can also be tested for the same MMR mutation?

In Abstract there is immunochemical instead of immunohistochemical.

In Introduction, first row, sentence started with "It is characterized by the presence of pathogenic mutations in DNA repair genes…", I would correct this sentence by naming of DNA mismatch (MMR) repair genes (MLH1, MSH2 etc), since many other DNA repair genes exists. 

In Paragraph 2.2, third row, sentence started with "In patients with loss of MLH1, methylation of MLH1 and/or somatic BRAF mutation status was analyzed." What does this sentence means, it is unclear and needs an improvement, for example what BRAF mutation status (V600E?). Rows 10 and 11, I would briefly characterize the analysis methods (without reader needs to read references 4 and 14).

In Paragraph 3.2, first row, "Regarding the analysis of family risk of developing CRC..." This sentence is very long and unclear for me, improvement is needed.

This whole article is based on LS and LLS pedigrees, so it would be nice to have an example pedigrees of (at least one) LS and LLS family.

Round 2

Reviewer 1 Report

The authors responded quite exhaustive to the questions raised in the last review.